# Fluorine-18 Labeled Urea-Based Ligands Targeting Prostate-Specific Membrane Antigen (PSMA) with Increased Tumor and Decreased Renal Uptake

**DOI:** 10.3390/ph15050597

**Published:** 2022-05-13

**Authors:** Falguni Basuli, Tim E. Phelps, Xiang Zhang, Carolyn C. Woodroofe, Jyoti Roy, Peter L. Choyke, Rolf E. Swenson, Elaine M. Jagoda

**Affiliations:** 1Chemistry and Synthesis Center, National Heart, Lung, and Blood Institute, National Institutes of Health, Bethesda, MD 20892, USA; xiang.zhang2@nih.gov (X.Z.); carolyn.woodroofe@nih.gov (C.C.W.); rolf.swenson@nih.gov (R.E.S.); 2Molecular Imaging Branch, National Cancer Institute, Bethesda, MD 20892, USA; tim.phelps@nih.gov (T.E.P.); jyotiroy14@gmail.com (J.R.); pchoyke@mail.nih.gov (P.L.C.); ejagoda@mail.nih.gov (E.M.J.)

**Keywords:** fluorine-18, PET, oxime, PSMA, lipophilicity, biodistribution

## Abstract

High expression of prostate-specific membrane antigen (PSMA) in prostate cancers prompted the development of the PSMA-targeted PET-imaging agent [^18^F]DCFPyL, which was recently approved by the FDA. Fluorine-18-labeled Lys–Urea–Glu-based oxime derivatives of [^18^F]DCFPyL were prepared for the comparison of their in vitro and in vivo properties to potentially improve kidney clearance and tumor targeting. The oxime radiotracers were produced by condensation of an aminooxy functionalized PSMA-inhibitor Lys–Urea–Glu scaffold with fluorine-18-labeled aldehydes. The radiochemical yields were between 15–42% (decay uncorrected) in 50–60 min. In vitro saturation and competition binding assays with human prostate cancer cells transfected with PSMA, PC3(+), indicated similar high nM binding affinities to PSMA for all radiotracers. In vivo biodistribution studies with positive control PC3(+) tumor xenografts showed that the kidneys had the highest uptake followed by tumors at 60 min. The PC3(+) tumor uptake was blocked with non-radioactive DCFPyL, and PC3(−) tumor xenograft (negative control) tumor uptake was negligible indicating that PSMA targeting was preserved. The most lipophilic tracer, [^18^F]**2a**, displayed comparable tumor-targeting to [^18^F]DCFPyL and a desirable alteration in pharmacokinetics and metabolism, resulting in significantly lower kidney uptake with a shift towards hepatobiliary clearance and increased liver uptake.

## 1. Introduction

Prostate cancer (PC) is the most common malignancy in men in the United States and Europe [1,2,3]. In recent decades, prostate cancer survival rates have improved; however, it is still a significant cause of death. Local PC is usually diagnosed with screening for prostate serum antigen (PSA serum testing), clinical examination, and imaging such as magnetic resonance imaging (MRI) followed by a biopsy of the prostate. Advanced PC, however, is commonly staged with computed tomography (CT), bone scans and positron emission tomography (PET), frequently using prostate-specific membrane antigen (PSMA)-targeted radioligands. Due to the higher sensitivity of PET over the other techniques, it is becoming more widely accepted as a diagnostic approach to identify sites of extra-prostatic disease. The metabolic radiotracer, 2-deoxy-2-[^18^F]fluoro-D-glucose, [^18^F]FDG, although commonly used in other cancers, has proven less useful in PC [4,5]. Carbon-11 or fluorine-18-labeled choline PET/CT showed promising results for the detection of bone metastases. However, these agents have limitations in terms of sensitivity and specificity [6]. This unmet clinical need led to the development of another class of radiotracers targeting the transmembrane protein PSMA, which is expressed in approximately 95% of PC cases including both primary and metastatic disease [7,8,9]. PSMA is a cell surface glycoprotein with carboxypeptidase and folate hydrolase enzymatic activities that has emerged as an important biomarker for PC and prompted the development of small-molecule inhibitors [10,11,12]. These smallmolecule inhibitors have proven to be suitable platforms for PET imaging with faster clearance rates and lower backgrounds.

The gallium-68 labeled PET tracer, Glu-NH-CO-Lys-(Ahx)-[^68^Ga]Ga-N,N′-Bis(2-hydroxy-5-(ethylene-betacarboxy)benzyl)ethylenediamine N,N′-diacetic acid ([^68^Ga]Ga-PSMA-11 (also named [^68^Ga]Ga-PSMA-HBED-CC)), is the most widely studied PSMA radiotracer [13,14,15,16,17]. It was first reported by Eder et al. in 2012 [18]. The initial clinical PET imaging study with this tracer demonstrated a significant advantage compared to conventional imaging used for the detection of recurrent PC [13]. [^68^Ga]Ga-PSMA-11 was recently approved by the Food and Drug Administration (FDA) for PET imaging of PSMA-positive lesions in men with prostate cancer [19]. However, the longer half-life of fluorine-18 (110 min) compared to gallium-68 (68 min) enables sufficient time for central production and local distribution of the tracers which is more pragmatic for most medical facilities. The extended imaging time with fluorine-18-labeled PSMA radiotracers may further increase the overall detection rate in patients with PC [20,21]. Moreover, fluorine-18 offers comparatively lower positron energy (fluorine-18, 633 keV vs. gallium-68, 1899 keV) with a resultant shorter positron range in the tissue, which may also improve image resolution [22,23]. Thus, the growing demand for PSMA-targeted PET imaging is likely to be better met by fluorine-18labeled radiotracers. Recently, Gust et al. proposed a molecular absorption spectrometry (MAS) method that uses fluorination as tool to improve bioanalytical labeling and suggested it as a potential alternative to ^18^F-PET [24].

A variety of fluorine-18labeled PSMA-targeted PET radiotracers have been developed for PC imaging [25,26,27,28,29]. The most extensively studied tracers of these classes are urea-based small molecule inhibitors, e.g., N-[N-[(S)-1,3-dicarboxypropyl]carbamoyl]-4-[^18^F]fluorobenzyl-L-cysteine ([^18^F] DCFBC), 2-(3-(1-carboxy-5-[(6-[^18^F]fluoropyridine-3-carbonyl)-amino]-pentyl)-ureido)-pentanedioic acid ([^18^F] DCFPyL), Glu-NH-CO-Lys-(Ahx)-[^18^F]AlF-N,N′-Bis(2-hydroxy-5-(ethylene-betacarboxy)benzyl)ethylenediamine N,N′-diacetic acid ([^18^F]-PSMA-11), and (2S)-2-[[(1S)-1-carboxy-5-[[(2S)-2-[[4-[[[(2S)-4-carboxy-2-[[(2S)-4-carboxy-2-[(6-[^18^F]fluoranylpyridine-3-carbonyl)amino]butanoyl]amino]butanoyl]amino]methyl]benzoyl]amino]-3-naphthalen-2-ylpropanoyl]amino]pentyl]carbamoylamino]pentanedioic acid ([^18^F] PSMA-1007) [30,31,32,33]. The clinical studies with the first-generation PSMA ligand [^18^F]DCFBC demonstrated slow clearance with high background activity [34]. The second-generation ligands, [^18^F] DCFPyL and [^18^F] PSMA-1007 showed high tumor: background ratios and favorable pharmacokinetics compared to other small molecules [31,35,36,37,38]. [^18^F]-DCFPyL was approved by the FDA in 2021 for the detection of possible early metastatic PC involvement [39]. A wide range of prosthetic groups and linkers have been introduced to improve pharmacokinetics and detection rates with PSMA PET [18,40,41,42,43]. These studies demonstrated favorable binding properties for more lipophilic compounds and inspired us to develop oxime derivatives with increased lipophilicity (Figure 1). Herein, we report the synthesis of the precursor, radiolabeling, and biological evaluation of these oxime derivatives in comparison with previously reported tracers [^18^F]DCFPyL and [^18^F]**1a**. The biological evaluations include in vitro binding studies to assess the affinity (K_d_) of these compounds for PSMA and in vivo biodistribution studies with PSMA-positive tumor mouse models to determine tumor targeting and differences in pharmacokinetics and metabolism.

## 2. Results

### 2.1. Radiochemistry

Oxime formation of the aminooxy functionalized lysine-urea-glutamate scaffold (**1** and **2**, Figure 1) with fluorine-18labeled aldehydes produced radiotracers [^18^F]**1b,** [^18^F]**2a** and [^18^F]**2b**. The radiosyntheses of the tracers were performed either manually ([^18^F]**1a** and [^18^F]**2a**) or by automated synthesis method ([^18^F]**1b** and [^18^F]**2b**). For the automated synthesis method, an external three-way valve was added to V10 (Figure 1) on the GE Tracerlab module (GE FX-N Pro) to accomplish the Sep-Pak fluorination of 6-[^18^F]fluoronicotinaldehyde. The overall radiochemical yield (2 steps) of the synthesis was 15–42% (*n* >10, decay uncorrected) in a 50–60 min procedure. The radiochemical purity was >98% with a molar activity of 300–360 GBq/µmol. The identities of the products were confirmed by comparing their HPLC retention times with co-injected, authentic non-radioactive standards. A representative HPLC profile for compound [^18^F]**1b** is shown in Figure 2. The calculated logP values indicated the lipophilicity order of the tracers are [^18^F]**2a** (1.45) > [^18^F]**2b** (0.58) > [^18^F]**1a** (0.04) > [^18^F]**1b** (−0.83) > [^18^F]DCFPyL (−0.94).

### 2.2. In Vitro Cell Binding Studies

All tracers exhibited high specific binding (B_sp_; 85–98%) with sub-nM affinity for PSMA using PC3(+) tumor membrane preparations (Figure 3A; Table 1). The K_d_ of [^18^F]DCFPyL (0.402 ± 0.121 nM, *n* = 5) was not significantly different from the K_d_ of [^18^F]**1a**–**b**, **2a**–**b** (Table 1) indicating that the addition of alkyl linkers to a fluorine-18labeled arene or heteroarene oxime moiety did not alter the binding affinity. A significant decrease of ~4-fold (*p* = 0.006) was observed in the K_i_ of [^18^F]**1a** (0.1 nM) compared to the K_i_ of [^18^F]DCFPyL (0.398 ± 0.055 nM, *n* = 3) suggesting that [^18^F]**1a** may have higher affinity than [^18^F]DCFPyL (Figure 3B; Table 1). PC3(+) tumor membrane preparations exhibited high PSMA expression levels [B_max_ = 13.95 ± 1.60 fmol/µg of protein, *n* = 5] with [^18^F]DCFPyL which compared favorably with the B_max_ values determined from similar saturation assays with the other four tracers, [^18^F]**1a**–**b**, **2a**–**b**.

### 2.3. In Vivo Biodistribution

The biodistribution of [^18^F]DCFPyL was determined in nude mice bearing human prostate cancer tumors transfected with PSMA (PC3(+) xenografts) at 30, 60, 90 and 120 min post-injection (Figure 4A,B). [^18^F]DCFPyL distributed rapidly and cleared from the blood and non-target tissues except for the tumor over the 120 min time course (Figure 4A). The kidney exhibited the highest uptake (133%ID/g to 50%ID/g) at all time points and decreased by 65% from 15 to 120 min. All other tissue uptakes except tumor were > 30 fold lower than kidneys at all times indicating that [^18^F]DCFPyL is dominated by renal clearance, as expected from published results [31]. The next highest uptakes after the kidneys occurred in the PC3(+) tumor in which [^18^F]DCFPyL was highly retained from 15 (18.6%ID/g) to 120 min (20.8%ID/g; Figure 4A). The tumor tissue to muscle ratio (T:M) steadily increased over the time course with an 11-fold increase from 15 (24 T:M) to 120 min (260 T:M). These tumor T:M increases are reflective of an increased rate of clearance from the muscle while in tumors the majority of the radioactivity was retained (Figure 4B). The retention of [^18^F]DCFPyL in the tumor with an accompanying increase in tumor T:M over time would indicate high-affinity binding to PSMA. However, this was not the case with the salivary glands in which 92% of the radioactivity had been cleared at 120 min and T:M decreased 34% from 15 (2.7 T:M) to 120 min (2.0 T:M). This lack of retention of [^18^F]DCFPyL and the low T:M ratios suggest that salivary gland PSMA expression levels in mouse, known to be lower than in humans, are insufficient to render a meaningful biodistribution model. With this in mind, salivary glands were not included in further biodistributions with the ^18^F-labeled analogues [35].

Additional [^18^F]DCFPyL biodistribution studies at 60 min were carried out with PC3(−) xenografts and PC3(+) xenograft groups that received [^18^F]DCFPyL alone or a coinjection with a blocking dose of non-radioactive DCFPyL [1000×; 40 µg] (Figure 4C,D). The biodistributions of the PC3(−) xenografts were comparable to the PC3(+) xenografts for the blood and all tissues except for the PC3(−) tumors (0.3%ID/g) which represented <2% of the uptake observed in PC3(+) tumors (25.6%ID/g). In the blocking studies with the PC3(+) xenografts, the group receiving DCFPyL exhibited >50% reduction in blood and most tissue uptakes (%ID/g) compared to the [^18^F]DCFPyL- only group. T:Ms were calculated to take into account these alterations in the input function and metabolism caused by the DCFPyL blocking dose. The PC3(+) tumor T:M (50:1) of the blocked group was significantly decreased (60%) compared to the [^18^F]DCFPyL-only group (141:1 T:M; *p* < 0.0001). This blocking taken together with the lack of uptake in PC3(−) tumors would indicate that tumor uptake represents specific PSMA binding. Other significant decreases in T:M ratios occurred in the kidney (96%) and spleen (81%) compared to the [^18^F]DCFPyL-only group. These decreases most likely are not entirely attributable to PSMA specific binding but could be a result of the altered metabolism in the kidney or cross-reactivity with glutamate carboxypeptidase III (GPCIII) in the spleen, respectively [44]. The only increase in T:M occurred in the liver (27:1 T:M; 2.7 fold) compared to the [^18^F]DCFPyL only group which most likely is due to a shift from renal towards hepatobiliary metabolism.

Initially, the biodistributions of the tracers ([^18^F]**1a**–**b**, [^18^F]**2a**–**b**) were evaluated at 60 min in PC3(−) xenografts and PC3(+) xenograft groups with or without a blocking dose of DCFPyL for direct comparison to the reference compound [^18^F]DCFPyL (Figure 5A–C). For all the tracers the kidneys and tumors exhibited the highest uptakes as was observed with [^18^F]DCFPyL, although differences in the blood and other tissues were observed indicating some alterations in pharmacokinetics and metabolism (Figure 5A). Radioactivity in the blood of [^18^F]**1b** (0.309%ID/g; *p* = 0.005) and [^18^F]**2b** (0.377 %ID/g; *p* = 0.034) was decreased significantly (40% and 27%, respectively) compared to [^18^F]DCFPyL (0.519%ID/g) indicating faster blood clearance. Conversely, [^18^F]**2a** in the blood (0.8297%ID/g; *p* = 0.034) was significantly increased by 1.6-fold vs. [^18^F]DCFPyL. With these significant changes in the blood input function between [^18^F]DCFPyL and these analogues, T:Ms were determined to assess the differences in PSMA targeting and metabolism (Figure 5B; Table 2). For [^18^F]**1b,** PC3(+) tumor T:M (204:1 T:M) was significantly increased by 1.6 fold compared to [^18^F]DCFPyL (124:1 T:M), whereas [^18^F]**2a** (74:1 T:M) and [^18^F]**2b** (90:1 T:M) significantly decreased by 40% and 27%, respectively. The PC3(−) tumor T:Ms for all the analogues were <1.0 (Table 2) and comparable to the PC3(−) tumor T:M of [^18^F]DCFPyL (Figure 4D), indicating that the uptake in the PC3(+) tumor is reflective of PSMA expression levels. In the DCFPyL blocking studies, PC3(+) tumor T:Ms of all four analogues were decreased compared to the non-blocking groups, with significant decreases occurring in [^18^F]**2b** PC3(+) tumor T:M (85%; *p* < 0.0001) and [^18^F]**2a** PC3(+) tumor T:M (48%; *p* < 0.0008, Figure 5C). These results indicate that PSMA targeting has been preserved for all the analogues compared to [^18^F]DCFPyL with [^18^F]**1a** exhibiting the highest tumor T:M ratios. Although [^18^F]**1b** had improved PC3(+) tumor targeting the kidney T:M was increased 1.4 fold compared to [^18^F]DCFPyL. [^18^F]**2a** was the only analogue in which kidney T:M was significantly decreased (55%; *p* < 0.0001) compared to [^18^F]DCFPyL (Figure 5B). Liver T:Ms were increased by >2-fold for all four analogues compared to [^18^F]DCFPyL with the greatest increase (6.8-fold) observed with [^18^F]**2a**.

Further pharmacokinetic studies were performed with [^18^F]**2a** which had appropriate tumor targeting and lower kidney uptake relative to [^18^F]DCFPyL (Figure 6A–D). Overall [^18^F]**2a** cleared rapidly from the blood to the kidneys which had the highest uptakes at all time points (Figure 6A). The tumor uptakes (%ID/g) were the next highest at the later times of 60 and 120 min. The liver uptake of [^18^F]**2a** was higher at the earlier times of 15 and 30 min whereas for [^18^F]DCFPyL the tumor uptakes were the next highest at all times (Figure 6B). [^18^F]**2a** was highly retained in the tumors (%ID/g) and tumor retention significantly increased (1.4-fold) from 15 to 120 min. Similarly, tumor T:Ms increased ~14-fold from 15 to 120 min indicating tumor retention and muscle clearance (Figure 6D). [^18^F]**2a** tumor T:M was comparable to [^18^F]DCFPyL at 15 and 120 min but decreased ~2-fold at 30 and 60 min compared to the [^18^F]DCFPyL tumor T:M. These modest decreases in [^18^F]**2a** tumor T:Ms most likely are attributable to changes in the blood input function and alterations in the kidney and liver metabolism compared to [^18^F]DCFPyL. [^18^F]**2a** blood radioactivity content (%ID/g) was higher at all time points compared to [^18^F]DCFPyL indicating slower rates of clearance of [^18^F]**2a** from the blood and other non-target tissues (Figure 6A,C). Further differences were observed in clearance of [^18^F]**2a** to the kidney (%ID/g; T:M) which significantly decreased (40% to 54%) at all time points except 120 min compared to [^18^F]DCFPyL while liver uptake (%ID/g; T:M) increased significantly (7 to 15-fold) over the time course (Figure 6B,D). Femur uptakes (%ID/g; T:M) of [^18^F]**2a** and [^18^F]DCFPyL were comparably low (<1%ID/g: <1.8 T:M) at all times except at 120 min in which [^18^F]DCFPyL femur T:M was 1.8 fold greater than [^18^F]**2a**, suggesting insignificant in vivo defluorination.

Since differences were observed in metabolism between [^18^F]DCFPyL and the other tracers, the fraction of radioactivity that represented intact tracer (%Parent) in blood was determined at 60 min by TLC (Table 2). Compared to [^18^F]DCFPyL and the other analogues, [^18^F]**2a** exhibited the greatest in vivo blood stability at 60 min with 85% (parent) of the total blood radioactivity remaining intact. Additional TLC analysis was performed with [^18^F]**2a** and [^18^F]DCFPyL to determine the fraction of parent remaining over the time course from 15 to 120 min in blood and kidneys. The % of parent [^18^F]**2a** in blood was relatively constant from 15 to 120 min (85 to 83% parent) which was greater than [^18^F]DCFPyL(50 to 40% parent) over the same time period. In contrast, the majority of radioactivity in the kidney was metabolites of both [^18^F]**2a**, and [^18^F]DCFPyL, however, the % of parent [^18^F]**2a** (29 to 15%), was greater than the % of parent [^18^F]DCFPyL(16 to 7%) at all time points.

In Table 3, [^18^F]**2a** and [^18^F]DCFPyL uptake (%ID/g) in the blood and kidney before and after correction for metabolites (parent) were compared over the 15 to 120 min time course. At all times parent [^18^F]**2a** in the blood was 3 to 7-fold greater than parent [^18^F]DCFPyL, indicating [^18^F]**2a** had increased stability in the blood and slower clearance. The kidney uptake (%ID/g) of the parent [^18^F]**2a** and [^18^F]DCFPyL was comparable over the time course except for 120 min in which [^18^F]**2a** was increased by 4-fold (Table 3). Parent [^18^F]**2a** kidney uptake was relatively unchanged over the time course indicating that this fraction of retained parent [^18^F]**2a** may be representative of specific binding to PSMA in the proximal tubules of the kidneys (Table 3) [44]. Similarly [^18^F]DCFPyL parent kidney uptake was retained from 15 to 60 min although a decrease was observed at 120 min.

### 2.4. PET Imaging Studies

Small animal PET imaging studies were performed with PC3(+) tumor xenograft mice at 60 min post-injection of [^18^F]DCFPyL or [^18^F]**2a** [3.7 to 7.4 MBq (100–200 µCi)] (Figure 7). Tumors and kidneys were easily visualized from PET images of xenografts injected with both [^18^F]DCFPyL and [^18^F]**2a** whereas the liver was only apparent in the [^18^F]**2a** image. From ROI analysis of the images the PC3(+) tumor uptakes of [^18^F]DCFPyL or [^18^F]**2a** were determined to be 19%ID/g and 14%ID/g, respectively, which were comparable to [^18^F]DCFPyL and [^18^F]**2a** tumor uptakes from the biodistribution studies. From similar ROI analysis the [^18^F]**2a** kidney uptake (98%ID/g) was reduced by ~50% compared to the [^18^F]DCFPyL kidney uptake (187%ID/g) whereas [^18^F]**2a** liver uptake (29%ID/g) was increased ~10-fold compared to the [^18^F]DCFPyL liver uptake (3%ID/g). These [^18^F]**2a** and [^18^F]DCFPyL quantitative imaging results were in agreement with the biodistribution results for the respective tissues.

## 3. Discussion

Recently, Bouvet et al. reported the influence of different prosthetic groups on PSMA-targeted radiotracers (DCFPyL analogues) with improved tumor uptake and clearance profile [41]. The highest tumor uptake in their study was achieved with the most lipophilic compound (**1a**, Figure 1), prepared via the oxime formation of 4-[^18^F]fluorobenzaldehyde with aminooxy precursor **1**. Moreover, they suggested that the low tumor uptake of the [^18^F]FDG linked oxime tracer could be due to the combination of high hydrophilicity and steric crowding [36]. This result inspired us to further investigate the effect of adding an alkyl chain between the PSMA-inhibitor lysine-urea-glutamate scaffold and labeled prosthetic groups to increase lipophilicity and decrease steric crowding of the labeled PSMA probe. Therefore, compounds [^18^F]**1b** and [^18^F]**2a**–**b** were designed with alkyl chains of various lengths, and an arene or heteroarene substituent.

The biological evaluation of all tracers found that PSMA targeting was preserved both in vitro and in vivo and for the most part was comparable to [^18^F]DCFPyL. In vitro, the labeled tracers and non-radioactive standards had retained specific and high nM binding to PSMA, with [^18^F]**1a** tending to have higher affinity than [^18^F]DCFPyL. All four analogues exhibited in vivo PSMA tumor-targeting comparable to [^18^F]DCFPyL with tumor uptakes and T:M ratios, ranging from 27 to 17%ID/g and 203 to 74 T:M, respectively, which were at least 8-fold greater than non-target tissues except for the kidney and liver. Tumor uptakes of [^18^F]**1a** tended to be higher than [^18^F]DCFPyL comparing favorably with previous findings [41]. Therefore, the modification of the canonical amide bond of DCFPyL to include an alkyl chain and oxime-linked [^18^F]fluorobenzyl or -pyridinyl substituent minimally affected in vivo tumor targeting to PSMA, indicating that in human patients, all four [^18^F]DCFPyL analogues would be expected to identify PSMA expressing lesions as has been clinically observed with [^18^F]DCFPyL [45].

In human patients [^18^F]DCFPyL has demonstrated favorable dosimetry within acceptable limits for diagnostic PET tracers, however high accumulation and retention in the kidneys and salivary glands could limit use in other clinical applications such as radionuclide therapies [35,46]. In a retrospective clinical trial Barber et.al reported 25% renal injury in patients treated with [^177^Lu]Lu-PSMA-617 and currently sufficient kidney function is an important criterion for patient eligibility for this recently FDA-approved therapy [47,48,49]. In addition, retrospectively, xerostomia was found in 24% of patients by Heck et.al. which most likely is an underreported adverse effect resulting from high PSMA expression levels in the salivary glands [50]. The uptake of PSMA targeted imaging agents in human salivary glands is specific and consistent with known high PSMA expression levels which is not the case for mice. The mouse PSMA, a homolog of human PSMA, has a 12-fold lower expression level in salivary glands with higher levels of non-specific binding and therefore, may not be as reliable to detect changes in specific PSMA uptake [51,52]. In contrast, the elevated kidney uptake observed in mice is comparable to humans representing both specific PSMA binding in the renal cortex and non-specific radioactivity in the urinary tract due to excretion [44,53]. In studies investigating other fluorine-18labeled PSMA inhibitors the physiochemical properties of the labeled prosthetic groups were found to affect the biological clearance profiles, therefore, modification of the ^18^F-labeled prosthetic group of [^18^F]DCFPyL may offer a strategy to lower kidney uptake [41]. In these pre-clinical studies only [^18^F]**2a** displayed a desirable alteration in pharmacokinetics and metabolism resulting in greater in vivo stability and significantly lower kidney uptake with higher liver uptake compared to [^18^F]DCFPyL. [^18^F]**2a** liver uptake significantly decreased over the 2 h time course whereas kidney uptake remained unchanged. This shift to hepatobiliary clearance by [^18^F]**2a** may, in part, be explained by an increase in lipophilicity (logP = 1.45) compared to the other analogues. It is interesting to note that the other analogue which had the next highest logP (0.58), [^18^F]**2b**, had higher liver uptake as observed with [^18^F]**2a** but comparable kidney uptake to [^18^F]DCFPyL. Similarly, liver uptake was increased with [^18^F]**1a** compared to [^18^F]DCFPyL, as reported by Bouvet et al. [41]. The rank order of the liver uptake (%ID/g) of all the tracers indicated a switch to hepatobiliary clearance that corresponded to the tracers logP rank order suggesting that the lipophilicity plays a role in determining the clearance profile of the tracer. These results suggest that [^18^F]**2a** may offer an alternative PSMA-targeting agent with decreased renal clearance in clinical applications.

## 4. Materials and Methods

The aminooxy precursor, (((S)-5-(((aminooxy)carbonyl)amino)-1-carboxypentyl)carbamoyl)-L-glutamic acid (**1**), non-radioactive standards, (((S)-5-(2-(aminooxy)acetamido)-1-carboxypentyl)carbamoyl)-L-glutamic acid 4-fluorobenzaldehyde oxime (**1a**) and (((S)-5-(2-(aminooxy)acetamido)-1-carboxypentyl)carbamoyl)-L-glutamic acid 6-fluoronicotinaldehyde oxime (**1b**), were prepared according to literature methods [41]. Fluorine-18 radiolabeled DCFPyL, 4-[^18^F]fluorobenzaldehyde, and 6-[^18^F]fluoronicotinaldehyde were prepared following a recently published method [54,55]. PBS 1X buffer (Gibco) was obtained from Life Technologies (Carlsbad, CA, USA). All other chemicals and solvents were received from Sigma-Aldrich (St. Louis, MO, USA) and used without further purification. Fluorine-18 in target water was obtained from the National Institutes of Health cyclotron facility (Bethesda, MD, USA). Chromafix 30-PS-HCO_3_ anion-exchange cartridges were purchased from Macherey-Nagel (Düren, Germany). Columns and all other Sep-Pak cartridges used in this synthesis were obtained from Agilent Technologies (Santa Clara, CA, USA) and Waters (Milford, MA, USA), respectively. Oasis MCX Plus cartridges were conditioned with 5 mL anhydrous acetonitrile. The thin-layer chromatography (TLC) plates for phosphorimaging were obtained from Miles Scientific (Newark, DE, USA). Phosphorimaging plates were read using a Fuji FLA5100 and the data was analyzed using Image Gauge V4.0. Flash chromatography was performed on a Teledyne Isco Combiflash Rf+ instrument using hexane:ethyl acetate gradients. NMR spectra were obtained on a 400 MHz Varian NMR and processed using MestReNova software. LC/MS data for small molecules were acquired on an Agilent Technologies 1290 Infinity HPLC system using a 6130 quadrupole LC/MS detector and a Poroshell 120 SB-C18 2.7 um column (4.6 × 50 mm). HRMS data were acquired on a Waters XEVO G2-XS QTOF running MassLynx version 4.1. Semi-prep HPLC purification and analytical HPLC for radiochemical work were performed on an Agilent 1200 Series instrument equipped with multi-wavelength detectors.

### 4.1. Precursor and Non-Radioactive Standard

#### 4.1.1. (((. S)-5-(6-(Aminooxy)hexanamido)-1-carboxypentyl)carbamoyl)-L-glutamic acid (2)

di-*tert*-Butyl-(((S)-6-amino-1-(*tert*-butoxy)-1-oxohexan-2-yl)carbamoyl)-L-glutamate (1.00 g, 2.05 mmol) was combined with 6-(N-*tert*-butyloxycarbonyl)aminooxyhexanoic acid (500 mg, 2.02 mmol) and triethylamine (0.42 mL, 3 mmol) in 30 mL of dichloromethane [56]. HBTU (1.15 g, 3.03 mmol) was added, and the mixture was allowed to stir overnight. The reaction was diluted with dichloromethane, then washed sequentially with 50 mL each of saturated NaHCO_3_, 1 N HCl, and water. The organic layer was dried over anhydrous Na_2_SO_4_ and evaporated under reduced pressure. The resulting residue was purified by flash chromatography with a gradient from 50 to 100% ethyl acetate in hexanes to yield 1.00 g (68%) of a colorless residue. ^1^H NMR (CDCl_3_): δ 7.64 (s, 1H), 6.37 (s, 1H), 5.33 (br s, *J* = 27.3 Hz, 2H), 4.31 (ddd, *J* = 19.4, 8.4, 4.6 Hz, 2H), 4.12 (q, *J* = 7.2 Hz, 2H), 3.88 (t, *J* = 6.4 Hz, 2H), 3.38 (m, 2H), 3.12 (m, 2H), 2.81 (s, 6H), 2.38–2.30 (m, 2H), 2.25 (t, *J* = 8.0, 2H); 2.04 (s, 2H), 1.89–1.51 (m, 10H), 1.47 (s, 9H); 1.46 (s, 9H), 1.45 (s, 9H); 1.44 (s, 9H); 1.26 (t, *J* = 7.1 Hz, 2H). MS: Calculated for C_35_H_65_N_4_O_11_ (M+H): 717.5; found 717.5.

The intermediate, tri-*tert*-butyl-(((S)-5-(6-((t-butoxycarbonyl)aminooxy)hexanamido)-1-carboxypentyl)carbamoyl)-L-glutamic acid was dissolved in 2 mL of dichloromethane. TIPSH (0.2 mL), followed by TFA (2 mL) were added. The reaction was stirred for 3 h at RT, then concentrated under reduced pressure. The resulting residue was dissolved in water and purified by reverse phase HPLC using a semi-preparative HPLC column and gradient mobile phase from 0–30% acetonitrile in water. Both phases contained 0.05% TFA. The desired product (**2**) was isolated as a white solid (TFA salt, 495 mg, 65%) after lyophilizing the relevant fractions. ^1^H NMR (400 MHz, Methanol-*d*_4_) δ 4.31 (dd, *J* = 8.6, 5.1 Hz, 1H), 4.25 (dd, *J* = 8.5, 4.9 Hz, 1H), 4.02 (t, *J* = 6.4 Hz, 2H), 3.18 (t, *J* = 6.8 Hz, 2H), 2.50–2.33 (m, 2H), 2.24–2.08 (m, 3H), 1.96–1.78 (m, 2H), 1.76–1.59 (m, 5H), 1.59–1.48 (m, 2H), 1.48–1.38 (m, 4H). HRMS: Calculated for C_18_H_33_N_4_O_9_ (M+H): 449.2248, found 449.2239.

#### 4.1.2. (((. S)-5-(6-(Aminooxy)hexanamido)-1-carboxypentyl)carbamoyl)-L-glutamic acid 4-fluorobenzaldehyde oxime (2a)

Compound 2 (20 mg, 36.7 µmol) was dissolved in 2 mL of MeOH, triethylamine (50 µL) and 4-benzaldehyde (11.1 mg, 89.2 µmol) were added. The reaction was stirred for 2 h at RT, then concentrated under reduced pressure. The desired major product was isolated by preparative HPLC (10–70% MeCN in H_2_O) followed by lyophilization of the relevant fractions (white solid, 15.4 mg, 62% yield). ^1^H NMR (400 MHz, Methanol-d4) δ 8.08 (s, 1H), 7.67–7.58 (m, 2H), 7.18–7.06 (m, 2H), 4.31 (dd, *J* = 8.6, 5.0 Hz, 1H), 4.26 (dd, *J* = 8.4, 4.9 Hz, 1H), 4.14 (t, *J* = 6.5 Hz, 2H), 3.16 (t, *J* = 6.8 Hz, 2H), 2.50–2.32 (m, 2H), 2.24–2.08 (m, 3H), 1.96–1.79 (m, 2H), 1.79–1.68 (m, 2H), 1.68–1.60 (m, 3H), 1.55–1.51 (m, 2H), 1.51–1.38 (m, 4H). HRMS: Calculated for C_25_H_36_FN_4_O_9_ (M+H): 555.2466, found 555.2463.

#### 4.1.3. (((. S)-5-(6-(Aminooxy)hexanamido)-1-carboxypentyl)carbamoyl)-L-glutamic acid 6-fluoronicotinaldehyde oxime (2b)

Compound **2** (20 mg, 36.7 µmol) was dissolved in 2 mL of MeOH, and triethylamine (50 µL) and 6-fluoronicotinaldehyde (11.2 mg, 89.2 µmol) were added. The reaction was stirred for 2 h at RT, then concentrated under reduced pressure. The desired major product was isolated by preparative HPLC (10–70% MeCN in H_2_O) followed by lyophilization of the relevant fractions (white solid, 11.4 mg, 46% yield).

^1^H NMR (400 MHz, Methanol-*d*_4_) δ 8.36 (d, *J* = 2.5 Hz, 1H), 8.26–8.19 (m, 1H), 8.17 (d, *J* = 0.5 Hz, 1H), 7.13–7.07 (m, 1H), 4.31 (dd, *J* = 8.6, 5.0 Hz, 1H), 4.26 (dd, *J* = 8.3, 4.8 Hz, 1H), 4.18 (t, *J* = 6.5 Hz, 2H), 3.20–3.13 (m, 2H), 2.41 (ddd, *J* = 8.5, 6.8, 3.5 Hz, 2H), 2.24–2.08 (m, 3H), 1.96–1.78 (m, 2H), 1.77–1.62 (m, 5H), 1.58–1.49 (m, 2H), 1.47–1.39 (m, 4H). HRMS: Calculated for C_24_H_35_FN_5_O_9_ (M+H): 556.2419, found 556.2418.

### 4.2. Radiochemical Syntheses

All radiochemical syntheses were performed according to the following two general procedures described below.

#### General Method

Procedure 1: Manual syntheses for compounds [^18^F]**1a** and [^18^F]**2a**

Fluorine-18 in target water (3700–7400 MBq) was diluted with 2 mL water and passed through an anion-exchange cartridge (Chromafix 30-PS-HCO_3_). The cartridge was washed with anhydrous acetonitrile (6 mL) and dried for 1 min. The [^18^F]fluoride from the cartridge was slowly eluted (0.5 mL/min) with its 4-formyl-N,N,N-trimethylbenzenaminium triflate precursor (5–7 mg) in 0.5 mL 1:4 acetonitrile: *t*-butanol. The Sep-Pak was further eluted with 0.5 mL acetonitrile and the eluent was collected in the same vial. The reaction mixture was heated at 120 °C for 2 min to produce 4-[^18^F]fluorobenzaldehyde. The radiolabeled intermediate was purified by passing the reaction mixture through a pre-conditioned Oasis MCX Plus cartridge and collected in a vial containing aminooxy precursor **1** or **2** (5 mg) in 0.2 mL water. The cartridge was flushed with 1 mL acetonitrile and the eluent was collected in the same vial. The solution was stirred for 10 min at 70 °C and the solvent was evaporated under N_2_ and reduced pressure. The HPLC buffer (3 mL) was added via a syringe. The mixture was injected into the HPLC for purification. The collected product was buffered to pH ~7 with 45 mM sodium phosphate. The identity and purity of the product were confirmed by analytical HPLC.

Procedure 2: Automated syntheses for compounds [^18^F]**1b** and [^18^F]**2b** on a GE Tracerlab FX-N Pro module

Fluorine-18 in target water (3700–7400 MBq) was diluted with 2 mL water and passed through an anion-exchange cartridge (Chromafix 30-PS-HCO_3_) followed by anhydrous acetonitrile (6 mL) and the cartridge was dried for 3 min under vacuum. The [^18^F]fluoride from the Sep-Pak was eluted with 5-formyl-N,N,N-trimethylpyridin-2-aminium triflate precursor (5–7 mg) in 0.5 mL 1:4, acetonitrile: *t*-butanol (in a syringe) via an external three-way valve. The mixture was passed through a pre-conditioned Oasis MCX Plus cartridge (incorporated between V13 and Reactor 1). The cartridge was flushed with 1 mL acetonitrile through the external three-way valve and the eluent was collected in the same vial (Reactor 1). To this solution in Reactor 1 was added the aminooxy precursor, **1** or **2**, (5 mg) in water (0.5 mL) from Vial 3. The solution was stirred for 10 min at 70 °C and the solvent was then evaporated under N_2_ and vacuum. The HPLC buffer (3 mL) was added from Vial 4. The mixture was transferred to Tube 2 and injected into the HPLC for purification. The collected product was buffered to pH ~7 with 45 mM sodium phosphate. The identity and purity of the product were confirmed by analytical HPLC.

HPLC conditions for purification: Agilent Eclipse plus C18 column (9.4 × 250 mm, 10 µm), mobile phase: B = ethanol, A = 50 mM phosphoric acid, flow rate of 3.5 mL/min.

**HPLC conditions for analysis**: Agilent Eclipse plus C18 (4.6 × 150 mm, 3.5 µm), B = acetonitrile, A = 0.1 M aqueous ammonium formate pH adjusted to 3.5 with trifluoroacetic acid, flow rate of 1 mL/min.

[^18^F]**1a:** The radiochemical yield was 21–32% (uncorrected, *n* > 5) in 50 min with a molar activity of 300–330 GBq/µmol. HPLC conditions for purification: 30% B in A, t*_R_* = ~18 min. HPLC conditions for analysis: 20% B in A, t*_R_* = ~5 min.

[^18^F]**1b**: The radiochemical yield was 37–40% (uncorrected, *n* = 9) in 45 min with a molar activity of 300–360 GBq/µmol. HPLC conditions for purification: 20% B in A, t*_R_* = ~20 min. HPLC conditions for analysis: 15% B in A, t*_R_* = ~4 min.

[^18^F]**2a**: The radiochemical yield was 15–27% (uncorrected, *n* > 5) in 50 min with a molar activity of 300 -330 GBq/µmol. HPLC conditions for purification: 50% B in A, t*_R_* = ~22 min. HPLC conditions for analysis: 20% B in A, t*_R_* = ~9 min.

[^18^F]**2b**: The radiochemical yield was 36–42% (uncorrected, *n* = 9) in 45 min with a molar activity of 300–360 GBq/µmol. HPLC conditions for purification: 35% B in A, t*_R_* = ~15 min. HPLC conditions for analysis: 20% B in A, t*_R_* = ~7 min.

### 4.3. Lipophilicity

The lipophilicities of the molecules were determined by calculating the value of the partition coefficient (logP) using ChemDraw 2019. The logP values of the molecules, [^18^F]DCFPyL, [^18^F]**1a**, [^18^F]**1b,** [^18^F]**2a,** [^18^F]**2b** were −0.94, 0.04, −0.83, 1.45, 0.58, respectively.

### 4.4. Cell Lines and Human Tumor Xenograft Mouse Models

PC3(−) (wildtype human prostate cancer cell line, PSMA negative) and PC3(+) (transfected with human PSMA) were provided by Dr. Hisataka Kobayashi [55,57]. Cell lines were grown at 37 °C in 5% CO_2_ in RPMI-1640 supplemented with 10% FBS, 2 mM L-glutamine and Pen/Strep/Amphotericin B**.** PC3(+) and PC3(−) cell suspensions [2 × 10^6^ cells; PBS:Matrigel (70:30)] from in vitro cell culture were subcutaneously implanted (right shoulder) into athymic mice (Athymic NCr- nu/nu, Charles River Laboratory, 4 weeks old) for use as positive and negative controls, respectively, for in vitro or in vivo studies. When tumors reached the appropriate size (>100 mg) the xenograft mice were used for in vivo biodistributions and imaging studies or the tumors were excised and further processed to obtain membrane preparations for in vitro assays as described previously [58].

### 4.5. In Vitro Binding Studies

In vitro saturation studies were performed to determine binding affinities (K_d_) and PSMA expression levels (B_max_) using tumor membrane preparations from PC3(−) and PC3(+) PSMA xenografts (human prostate cancer cell line transfected with human PSMA; PC3(+)). A constant aliquot of the tumor membrane preparation was added to increasing concentrations of the tracers (0.5–70 nM) in duplicate (total bound activity (B_t_)); non-specific binding (B_nsp_) was determined by adding non-radioactive DCFPyL (10^−6^ M) to another set of duplicates. For competition assays a constant concentration of [^18^F]DCFPyL (0.5 to 1.0 nM) and increasing concentrations (0–1000 nM) of competitors (non-radioactive standards; DCFPyL, **1a**–**b**, or **2a**–**b**) were added to membrane aliquots. After incubation (2 h at RT) separation of bound [^18^F]DCFPyL from free was accomplished by filtration using GF/C filter papers followed by 2 washes with saline. Filter papers were collected, and the radioactive content was quantified by gamma counting (PerkinElmer 2480 Wizard3). From the saturation studies, the K_d_ and B_max_ were determined from 6- 8 concentrations of the radiolabeled tracers and analyzed using non-linear regression curve fitting (one-site specific binding hyperbola); from the competition studies, inhibitory constants (K_i_)’s were determined from 8–10 competitor concentrations of non-radioactive standard/DCFPyL [PRISM (version 7.0 Windows), GraphPad software, San Diego, CA]. Aliquots of each membrane preparation were taken for the determination of the protein concentration (Bradford method).

### 4.6. Biodistributions

Tumor-bearing mice (tumor weights: 0.1–0.8 g) were injected while awake via the tail vein with each of the tracers [0.74–3.7 MBq (20 to 100 µCi), 10 to 80 pmol] and euthanized (via CO_2_ inhalation) at selected times. For the blocking studies, mice were coinjected with one of the tracers [0.74–3.7 MBq (20 to 100 µCi), 10 to 80 pmol] + DCFPyL (1000×: ~10 to 80 nmol) and euthanized at 60 min post-injection. Blood samples and tissues were excised from each animal, weighed, and radioactivity content was determined (Perkin Elmer 2480 Wizard3). Radioactivity content in the blood and each tissue was expressed as % injected dose per gram of tissue [%ID/g; (Formula (1))] and then normalized for body weight to a 20 g mouse (Formula (2)) from which Tissue:Muscle ratios [T:M; (Formula (3))] were determined as follows:(1)$$\%\mathrm{ID}/g\;=\frac{[\mathrm{counts}\;\mathrm{per}\;\mathrm{minute}\;{\left(\mathrm{cpm}\right)}_{\mathrm{tissue}}\;/\;\mathrm{tissue}\;\mathrm{weight}\;\left(g\right)]\times 100}{{\mathrm{cpm}}_{\mathrm{total}\;\mathrm{injected}\;\mathrm{dose}}}$$


%ID/g (normalized to a 20 g mouse) = (%ID/g) × (body weight/20 g)(2)




(3)
$$T:M=\frac{\%\mathrm{ID}/g_{\mathrm{tissue}}}{\%\mathrm{ID}/g_{\mathrm{muscle}}}$$



Statistical analysis of the differences between the 2 groups was carried out using the Student’s *t*-test with *p* < 0.05 as significant (GraphPad In Stat 3 for Windows).

In some cases, additional blood samples and/or tissue samples after gamma counting were taken for determining the fraction of intact radiolabeled tracer (parent) using thin-layer chromatography (TLC). For these TLC determinations: tissues were placed in equal volumes of acetonitrile and homogenized, or serum was obtained from the blood samples and mixed with an equal volume of acetonitrile. Following centrifugation of the samples, supernatants were collected and the radioactive content of the supernatants and pellets were determined. The supernatants were then applied to thin-layer chromatography (TLC) plates. The TLC plates were developed [solvent system: ethyl acetate (80%), methanol (10%) and acetic acid (10%)], and exposed on a phosphorimaging plate which was scanned the next day.

### 4.7. PET Imaging Studies

Tumor-bearing mice were anesthetized with isoflurane/O_2_ (1.5–3% *v*/*v*) and imaged at various times after intravenous injection of each tracer [2.6 to 3.7 MBq (70 to 100 µCi)]. Whole body static PET images were obtained at 2 bed positions (FOV = 2.0 cm, total imaging time: 10 min) using a BioPET scanner (Bioscan Inc., Washington, DC, USA). The images were reconstructed by a 3-dimensional ordered-subsets expectation maximum (3D-OSEM).

## 5. Conclusions

Fluorine-18-labeled urea-based PSMA inhibitors were prepared either manually or automatically in high radiochemical yield using the prosthetic group 4-[^18^F]fluorobenzaldehyde or 6-[^18^F]fluoronicotinaldehyde. [^18^F]**2a** displayed a desirable alteration in pharmacokinetics and metabolism resulting in significantly lowering the kidney uptake while maintaining high-affinity binding to PSMA compared to [^18^F]DCFPyL. Therefore, [^18^F]**2a** may be of use in clinical applications to reduce the radioactive dose to kidneys while maintaining high tumor uptake.

## Data Availability

Data is contained within the article.

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
