# Peer review of "Fluorine-18 Labeled Urea-Based Ligands Targeting Prostate-Specific Membrane Antigen (PSMA) with Increased Tumor and Decreased Renal Uptake"

_pharmaceuticals, 2022, doi:10.3390/ph15050597_

Round 1

Reviewer 1 Report

Given the high incidence, morbidity and mortality of prostate cancer, it is clear that any advance in diagnosis or treatment is of high interest and must be published. The present paper is such a paper dealing with the development of better radiopharmaceuticals. As presented, with F-18 as the workhorse, the primary focus is on diagnostic use, with the potential of lowering side effects and perhaps better detection through improved biodistribution. However the manuscript may also have implications for future development of better radionuclide therapy agents against PC.

The presented work represents a substantial piece of systematic work, giving some incremental gains in both drug design and fundamental understanding the biological background for the observed  biodistriutions. The manuscript is condensed, full of facts, and almost flawless. It deserves publication more or less as is. The whole bulk of data included in the paper must be included, as it any small piece of this may some day provide a clue to a new important pharmaceutical.

During close review , I have found a number of minor things that should/could  be corrected in the final version:

In general: The manuscript just talks about "xenografts" , but the meaning must be "nude mice with xenograft human PC tumor lines". I am not sure even the animal species is ever mentioned. At least this reviewer was still a bit in doubt when arriving at line 118.

Line 40: ...prostate cancer --> prostate cancer cases.

Figure 3 B, ordinate axis label and text is difficult to read.

Table 1.  "1Kd " and "2Ki" must be written more clear, so it means  Kd and Ki with reference to notes 1) and 2) below the table .

Line 133: with this, in mind.... ---> with this in mind

Table 2, last row, column 1 "% Parent in blood"-->%Parent compound in blood"

Figure 7, caption: 

Activity of both injected compounds should be stated, both in Bq and (as option) in uCi in paranthesis  ( as done later in paper).

Ar the colorscales abritary units, or can the two images be numerically compared?

Line 266 "FDG linked oxime tracer"... where does that FDG come from? I can not understand the context.

Line 456 and also in line 457: missing ) after uCi

Lines 460-461   (1)... (2) and (3) refer to formulas 1, 2 and 3 just below, but the naive reader ( like this reviewer thought it lokked like reference numbers, and that gave me no clue.

Line 467, formula 3: I think it should be "tissue / muscle" not the more specific "tumor/muscle"

Line 478 "and exposed to a phosphorimager plate"---> and exposed on a phosphorimager plate.

Reference 43,.. is that a book/monograph? Then please give some more details so it can be found. (ISBN , publisher ..)

Reference 44, the doi designation apears double, and written with the hypertext link ...

Reviewer 2 Report

This manuscript explores the potential efficacy of oxime derivatives of [18F]DCFPyL as PMSA-targeted imaging agents. The authors characterize  in vitro binding and murine biodistribution of a series of analogs, and identify one candidate (2b) that has lower kidney uptake and higher hepatobiliary clearance. [18F]DCFPyL and 2b are then compared in preliminary PET imaging studies in mice.

  1. The statistical methods used do not appear to be adequate. in Table 1 it appears that multiple comparisons are being made between the 4 novel radioligands and [18F]DCFPyL, but a simple t-test is used to compare each radioligand to [18F]DCFPyL. ANOVA followed by post-hoc testing should be employed. This also appears to be the case for other comparisons, e.g. Figure 5.
  2. While decreased kidney uptake and increased hepatobiliary clearance will reduce dosimetry to the patient's kidneys, the downside is that increased liver uptake may obscure metastases. is there evidence that kidney uptake may be limiting for the clinical use of PMSA radioligands?
  3. It would add to the manuscript to have quantitative imaging of a series of mice with with images taken around the 15, 30, 60, and 120 min windows and a figure showing quantitation of ROIs for the kidney, liver, and tumor for [18F]DCFPyL and 2b

Reviewer 3 Report

Prostate cancer is the most common malignancy in men in the developed countries. Tracers targeting the transmembrane protein PSMA, which is expressed in many prostate cancers can have a great importance in the diagnostic efforts against of this malignancy, and in the therapy also. Therefore, the selection of the topic has high relevance.

Moreover, the application of chemoselective reactions to conjugate the biological vectors with prosthetic groups is a very effective way for creating radioligands. The authors recently have developed a very attractive, cartridge-based synthesis of 6-[18F]fluoronicotinaldehyde, which can open new production strategies for such, heteroaromatic moiety-containing ligands. Oxime-formation can grant a bigger metabolic stability as well, if we compare it with the common amide-bond methods. Therefore, the work-concept represents a very high quality. Indeed, the research design seems to me appropriate, and the experimental methods are described adequately, but I have some doubt in the explanation of the results.

Authors wanted to tailor the lipophilicity, therefore they introduced a spacer between the Lysine-urea-Glutamate subunit and the prosthetic group. If the logP has a great importance, why they did not select more lipophilic ones to improve the differences between the analogues (line 300)?

The application of heteroaromatic prosthetic group instead of benzaldehyde will decrease the logP but offers other type of benefits in the synthesis (higher yield, easier automatization – line 406-). If the goal was to increase logP it would be advantageous to “prepare” the effect and overcompensate it with the use of lipophilic pharmacokinetic modifier, but using 6-[18F]fluoronicotinaldehyde instead of benzaldehyde-version.

It is strange for me the intention of decreasing the kidney uptake. The bioavailability of the radiopharmacons has great importance, especially in the case of therapeutic applications. But, if one operates with fluorine-labelled analogues, I can not see the therapeutic pair of this diagnostic tool. (It would be a different situation if the authors would develop radiometal-labelled theragnostic pairs.) If one tailors the excretion from the kidney to a mixed kidney/liver route – without to significantly improve the accumulation and retention of the targeted tissue - the only effect will be to cover the abdominal tract with a false signal, and the ligand will not be suitable to detect malignancies near to this aera (line 283).

There are some misspellings in the manuscript, such as the missing superscript in the “Table 1.”. I would suggest harmonizing the outlook and the size of the figures (for example 4 vs. 5), the 1st legend of the Table 3. is incorrect. In the 2.3. section the explanation is correct, but not easy to understand. Perhaps the application of more simple, shorter sentences would be beneficial for understanding.

Independently from these remarks I believe that in this manuscript one can read valuable research and a nice realisation of the experimental work.

Reviewer 4 Report

The manuscript entitled „Fluorine-18 Labeled Urea-Based ligands Targeting Prostate-Specific Membrane Antigen (PSMA) with Increased Tumor and Decreased Renal Uptake“ was submitted by F. Basuli and co-workers to the journal “Pharmaceuticals” to be considered for publication as an “Article”. The presented study reports the manually/automated synthesis of three precursor (two different lengths of spacers and phenyl or pyridyl- derivatives) compounds which were then radio-labelled and biologically investigated covering determinations of Kd and biodistribution (uptake studies, imaging). All in all, the 18F-labelled compound 2a showed to be most promising among the presented compounds compared to the lead [18F]DCFPyL, endorsing further studies. The manuscript is well written and gives a sound overview in respect to their research question. The study is generally worth publishing in the section “Radiopharmaceutical Sciences” of the journal “Pharmaceuticals”. However, there are some aspects, mainly formal issues, which are suggested to be addressed by the authors to reconsider this manuscript.

 “Prostate cancer (PC) is the most common malignancy in men in the United States and Europe [1-3].” The authors are kindly asked to provide newer studies. When looking at literature, it seems that newer publications than from 2010 (reference 3), from 2016 (reference 1) exist. As a suggestion for example doi: 10.14740/wjon1191 from 2019, doi: 10.22034/APJCP.2018.19.4.1041 from 2019, https://doi.org/10.1016/j.neo.2020.09.002 from 2020 or similar publications.

The authors used the PSMA-inhibitor Lys-Urea-Glu scaffold. However, I think the approach of the authors would be strengthen if the meaning of PSMA is pointed out more clearly. For such a pronunciation of the meaning of PSMA, the authors should add some few comments, considering recent literature such as: https://doi.org/10.1038/s41391-021-00394-5 from 2021, https://doi.org/10.3390/diagnostics11030552 from 2021, https://doi.org/10.2967/jnumed.120.257238 from 2021 for instance.

The length of the alkyl spacer nor the modifications in the oxime backbone did not influence the binding property of the compounds. However, if I did not miss it in the manuscript, what was the intention of the authors to vary the length of the spacer and the backbone moiety. There is only a notice in the discussion but not in the introduction. The authors are asked to comment on that earlier in the manuscript and provide reasons for this particular approach (i.e. n=5, pyridine=heteroaren in particular).

The authors synthesized the compounds and performed purification using preparative HPLC? Is this a common procedure at your lab or did the product contain too many impurities straight after synthesis?

The authors estimated the logP value by calculation using ChemDraw. However, it is always better to confirm that the software is in agreement with experimental determination of this value. How about evaluating the logP of one representative in order to proof the suitability of the applied software?

In the introduction, the authors discuss the impact of 18F-PET and carbon-11 or fluorine-18 labeled choline PET/CT as well as gallium-68 labeled PET, also considering different half-lifes and positron-energy, concluding that 18F-PET seems most suitable. However, limitations of this approach are not addressed but should be mentioned for the sake of completeness. The group of Gust et al. proposed a MAS-based method that uses fluorination as tool to improve bioanalytical labelling and suggested it as potential alternative to 18F-PET due to certain issues, also suited for uptake studies. This techniques seems very promising and the general applicability of this technique to the quantification of peptides was published most recently as continuation of their former work. The authors should embed such developments in their study to point out that F can be used as tracer but without radioactivity and without the restriction of half-lifes. However, the MAS-based method is still in its infancy but promising for the future. As a rebuttal, the authors could discuss that 18F-PET allows for imaging studies.

Please avoid page break with Figure 2A and 2B. This problem is probably solved after close editing by the publisher. Although there will be a close editing, the authors are asked to correct formal inconsistencies such as use of bold, introduction of every abbreviation at its first mention, justified text, etc.

Maybe it is possible to use colors in Figure 2B. Differently shaped/dotted curves in black and grey colors are hard to read and could be designed more reader-friendly.

In Figures 4, 5, the interception of the y-axis seems not perfectly suitable in each case as error bars are cut or since particular data are hard to read. Maybe the authors can consider a revision.

The authors are asked to include the “Institutional Review Board Statement” at the end of the article as suggested in the template, regarding their experiments using their mouse models.

Round 2

Reviewer 4 Report

The authors Falguni Basuli et al. submitted a revised version of their manuscript “Fluorine-18 Labeled Urea-Based ligands Targeting Prostate-Specific Membrane Antigen (PSMA) with Increased Tumor and Decreased Renal Uptake” to be considered for publication as a full article in the journal “Pharmaceuticals”.

They took into account all the previously mentioned concerns and suggestions – with one exception. The authors did not follow the suggestion to determine the logP value experimentally, but kept the approach of assessing it only in silico. However, the authors have refuted that the logP value was for estimation and therefore, in their opinion, no further experiments are needed. This is reasonable, since the result of their work is not worse without the experimental determination of the logP value. Otherwise, the authors have made the appropriate changes and improvements in their revised version.

There is one aspect that I actually wanted to mention in the first review, but unfortunately seemed to forget: Just recently, the FDA approved the first PSMA-targeted radiopharmaceutical. The inclusion of this brand new novelty in their manuscript would definitely show that the authors are researching on a hot topic! The following reference is recommended: 10.1038/d41573-022-00067-5. Perhaps they can include this in their addendum (lines 335-339)?

All in all, I am happy to see a nice manuscript that fits into the "Radiopharmaceutical Sciences" section of the journal and therefore support the further processing of the manuscript for publication. All the best and kind regards!

Author Response

The authors Falguni Basuli et al. submitted a revised version of their manuscript “Fluorine-18 Labeled Urea-Based ligands Targeting Prostate-Specific Membrane Antigen (PSMA) with Increased Tumor and Decreased Renal Uptake” to be considered for publication as a full article in the journal “Pharmaceuticals”.

They took into account all the previously mentioned concerns and suggestions – with one exception. The authors did not follow the suggestion to determine the logP value experimentally, but kept the approach of assessing it only in silico. However, the authors have refuted that the logP value was for estimation and therefore, in their opinion, no further experiments are needed. This is reasonable, since the result of their work is not worse without the experimental determination of the logP value. Otherwise, the authors have made the appropriate changes and improvements in their revised version.

There is one aspect that I actually wanted to mention in the first review, but unfortunately seemed to forget: Just recently, the FDA approved the first PSMA-targeted radiopharmaceutical. The inclusion of this brand new novelty in their manuscript would definitely show that the authors are researching on a hot topic! The following reference is recommended: 10.1038/d41573-022-00067-5. Perhaps they can include this in their addendum (lines 335-339)?

The reference has been included, line 305.

All in all, I am happy to see a nice manuscript that fits into the "Radiopharmaceutical Sciences" section of the journal and therefore support the further processing of the manuscript for publication. All the best and kind regards!